# Physical Activity, Physical Well-Being, and Psychological Well-Being: Associations with Life Satisfaction during the COVID-19 Pandemic among Early Childhood Educators

**DOI:** 10.3390/ijerph18189430

**Published:** 2021-09-07

**Authors:** Ken Randall, Timothy G. Ford, Kyong-Ah Kwon, Susan S. Sisson, Matthew R. Bice, Danae Dinkel, Jessica Tsotsoros

**Affiliations:** 1Department of Rehabilitation Sciences, College of Allied Health, University of Oklahoma Schusterman Center, 4502 East 41st Street, Tulsa, OK 74135, USA; Jessica-Tsotsoros@ouhsc.edu; 2Department of Educational Leadership and Policy Studies, Jeanine Rainbolt College of Education, University of Oklahoma Schusterman Center, 4502 East 41st Street, Tulsa, OK 74135, USA; tgford@ou.edu; 3Department of Instructional Leadership and Academic Curriculum, Jeanine Rainbolt College of Education, University of Oklahoma, 820 Van Vleet Oval, Norman, OK 73019, USA; kkwon@ou.edu; 4Department of Nutritional Sciences, College of Allied Health, University of Oklahoma Health Sciences Center, 1200 North Stonewall Avenue, Oklahoma City, OK 73104, USA; Susan-Sisson@ouhsc.edu; 5Department of Kinesiology and Sport Sciences, University of Nebraska at Kearney, 2504 9th Ave, Kearney, NE 68849, USA; bicemr@unk.edu; 6School of Health and Kinesiology, College of Education, Health, and Human Sciences, University of Nebraska at Omaha, H & K Building, 6001 Dodge Street, Omaha, NE 68182, USA; dmdinkel@unomaha.edu

**Keywords:** physical activity, physical well-being, psychological well-being, life satisfaction, COVID-19 pandemic, early childhood educators

## Abstract

Seeking personal well-being and life satisfaction during a global pandemic can be daunting, such is the case for early care and education teachers who were considered non-health care essential workers during the COVID-19 pandemic. The potential changes in their physical activity, along with their overall physical and psychological well-being, may have ultimately influenced their life satisfaction. These changes included the potential for increased sedentary behaviors. Despite the high health risks associated with these factors during the pandemic, the role of physical activity in early care and education teachers’ well-being and life satisfaction remains largely unknown. The purpose of this study is to examine the associations of physical activity and sedentary behaviors with teacher well-being and life satisfaction during the COVID-19 pandemic. In doing so, we explored two competing models of the relationship between the teachers’ physical activity, well-being, and life satisfaction, one with physical activity as a mediator and the other with teachers’ well-being as a mediator. An online survey, that collected information on physical, psychological, and professional well-being, job demands, and life satisfaction, was completed by 1434 US ECE teachers in 46 states. To test our hypothesized models, we conducted confirmatory factor analyses, followed by structural equation modeling. Of the respondents, 77% were overweight or obese and only 39% met the recommended 150 min of moderate physical activity per week. They had a mean life satisfaction score that qualifies as slight satisfaction, they experience moderate stress, and, collectively, are approaching the threshold for depression yet still reflect moderate-to-high work commitment. The empirical test of our competing mediation models found the model where teacher well-being mediated the association between physical activity, sedentary behavior, and life satisfaction was the superior model. The relationships between physical activity, sedentary behavior, and overall well-being suggest that these modifiable risk factors can be addressed such that early care and education teachers can improve their overall physical and psychological well-being, along with their life satisfaction.

## 1. Introduction

Seeking personal well-being and life satisfaction can be challenging endeavors even in the best of times. Adding a global pandemic to this set of challenges can make these pursuits even more daunting, particularly as it relates to adjusting to changes in both work and personal life. For some, the changes associated with the COVID-19 pandemic may have involved working from home, while for others it meant continuing work under very different conditions, still for others, it meant coping with the reality that their work sites had closed [1]. Early care and education (ECE) teachers serving children ages 0–5, like everyone else, had little time to make what were often significant adjustments to their lives and care for self. Moreover, early in the pandemic, the Centers for Disease Control and Prevention (CDC) categorized ECE teachers as frontline non-health care essential workers [2], which meant working under the extremely stressful and anxiety-inducing conditions that characterized the early months of the pandemic. As a result, the pandemic-induced changes experienced by ECE teachers likely influenced multiple aspects of their well-being, including physical and psychological well-being, which, by extension, likely influenced their overall life satisfaction.

Among the major concerns, public health experts raised during the pandemic were the likelihood of reduced physical activity (PA) and increased sedentary behaviors (SB) [3,4,5], which were more pronounced among women than men [3]. The evidence on the influence of PA and SB on physical and mental health outcomes and overall life satisfaction in the general population is well established [4,6]. This could be quite concerning for ECE teachers, who have been previously known to experience poor working conditions and poor physical and psychological well-being [7,8,9,10].

Despite the potentially high health risks associated with lack of PA and increased SB among ECE teachers during the pandemic, the role of PA and SB in their well-being and life satisfaction remains largely unknown. As such, the purpose of this study was to explore the associations of PA and SB with ECE teachers’ well-being and life satisfaction. We explore two different models of the relationship of PA and SB with well-being and ECE teachers’ life satisfaction: one with PA and SB as mediators and the other with teachers’ well-being as a mediator.

### 1.1. Benefit of Physical Activity for ECE Teachers

PA is conceptualized as any bodily movement produced by skeletal muscles that require energy expenditure, which can be expressed in metabolic equivalents (METs), with more-vigorous PA entailing higher MET levels [11,12]. SB is defined as any waking behavior with an energy expenditure of fewer than 1.5 METs and includes prolonged sitting and staying in a reclining posture [13]. PA reduces the risk of cardiovascular diseases, strokes, hypertension, type 2 diabetes, osteoporosis, obesity, certain types of cancer, anxiety, and depression [14,15]. Conversely, SB increases the risk for those same conditions [16,17,18]. PA is particularly important for ECE teachers, as previous research has found that those who were more physically active were less emotionally exhausted and had less intention to leave their current position [19]. Additionally, ECE teachers serve as role models to children by helping them develop positive behaviors and attitudes toward PA [20,21,22]. National PA recommendations within the United States have been established to guide individuals toward a healthier lifestyle [23]. PA recommendations for healthy adults include the accumulation of 150 min per week of moderate-intensity aerobic activity, 75 min per week of vigorous-intensity aerobic activity, or an equivalent combination of moderate- and vigorous-intensity aerobic activities [23]. However, only approximately 29% of ECE teachers in the United States meet the national recommendations for PA suggesting this is an important behavior to address [24].

### 1.2. The Importance of Well-Being and Life Satisfaction for Teachers in the COVID-19 Pandemic

Well-being is a multivariate concept characterized by physical and mental health, happiness, and overall satisfaction with life [25]. We [8,9] described the concept among ECE teachers as “whole teacher well-being,” which consists of physical, psychological, and professional well-being. Model of Whole Teacher Well-being [8] depicts this tripartite view of teacher well-being and its constituent elements. Physical well-being consists of the ability to perform activities and carry out social roles unhindered by physical limitations and experiences of bodily pain, it also includes medical status, weight in proportion to height (body mass index or BMI), sleep, and potential musculoskeletal issues [26]. Psychological well-being consists of positive emotions and feelings of happiness [27] and has numerous facets including perceptions of stress and resilience, or the ability to maintain or regain mental health despite experiencing adversity [28]. Professional well-being refers to people’s feelings about their work and job based on the definition of well-being from the Centers for Disease Control and Prevention [2]. This includes indicators such as self-efficacy and job commitment [9,29]. Life satisfaction is an overall subjective appraisal of an individual’s life experiences [30] and is strongly associated with mental health in adolescents and adults, with general health, gender, and income as contributing factors [31]. Quality of life is the confluence of the physical, psychological, and social domains of health within the context of an individual’s experiences, beliefs, and perceptions [32]. Life satisfaction and related quality of life are influenced by both physical and psychological well-being [33].

Although the pandemic has been a global event, it has had a somewhat disproportionate impact on certain groups of people, especially those classified as essential workers, such as health care providers and ECE teachers. During the pandemic, many ECE teachers risked COVID-19 exposure by continuing to care for children during the outbreak. However, these benevolent acts have gone under-acknowledged or appreciated as have these valued members of the teaching workforce. Other ECE facilities closed due to safety concerns, adding additional layers of life disruption and financial stress for teachers who are low-wage earners, use public assistance, and/or are persons of color [34].

Even before the pandemic, studies indicated that ECE teachers experience a multitude of challenges to their overall well-being due to job-related physical and psychological stresses [9,24,34,35,36,37,38]. Pre-pandemic psychological ill-being for ECE teachers was characterized by high levels of personal stress and depressive symptoms and far surpassed national averages [34,39,40,41,42]. They also reported substantial job-related stress [43,44] and burnout [45]. These trends are important because poor teacher physical and psychological well-being are negatively associated with their professional well-being and the quality of care they provide [9,46]. The disruptions and upheaval of a global pandemic only served to further complicate and disrupt these trends in ECE teachers’ well-being and quality of life. Studies of aspects of whole-teacher well-being examined various perspectives ranging from physical to psychological to professional, as well as their associations with teachers’ life satisfaction. Kern et al. [47] used regression analysis to estimate cross-sectional associations between self-reported physical health, life satisfaction, job satisfaction, and organizational commitment in a sample of school employees, 60% of whom were teachers. They determined that staff with higher engagement (defined as vigor and dedication to work) and better relationships reported greater job satisfaction and organizational commitment. Colomeischi [48] demonstrated a significant association with life satisfaction, engagement, and emotional intelligence via multiple linear regression in a sample of teachers in Romania. Further, Kardas and colleagues [49] reported a model in which life satisfaction, along with gratitude, optimism, and hope, accounted for approximately 51% of the variance of psychological well-being. Other studies focusing on psychological well-being reported that higher occupational stress was related to lower life satisfaction [50], and it was largely associated with psychological well-being, but not substantially related to indices of physical well-being [51]. Most of these studies suggest significant associations between various well-being indicators and life satisfaction, but the direction of, and mechanism to explain, these associations remain largely unknown. In addition, limited research evidence exists on the associations between various aspects of well-being and life satisfaction for ECE Teachers. As Hall-Kenyon and colleagues [29] note in their review of the literature related to well-being in ECE teachers, this literature is “deeply fragmented, rather narrow and limited.” They recommended research that attends more broadly to factors of well-being, including emotional and physical health and life satisfaction.

### 1.3. Physical Activity, Sedentary Behavior, and Overall Well-Being as Potential Mediators of Life Satisfaction

PA and SB play critical roles in various aspects of one’s well-being and life satisfaction. High PA and low SB are associated with higher life satisfaction and higher self-reported health [52]. PA plays a vital role in preventing numerous acute and chronic physical health conditions such as obesity, cardiovascular disease, cancer, and diabetes [23,53,54]. Moreover, PA can improve mental health and depression [55,56]. Parra-Rizo and Sanchis-Soler [57] found that older adults with high levels of PA had greater functional skills for activities of daily living that contribute to life satisfaction and overall well-being. Maher and colleagues [58] reported similar findings in that consistent practices of PA were positively associated with life satisfaction among middle and older adults, but not in young adults. Ginoux, et al. [59] found that regular PA improved both physical and psychological well-being, which included control over leisure time and relaxation. Nowak and colleagues [60] determined that the type of PA, including PA in the household, during transportation activities, and those related to occupation, had positive associations with quality of life but not with overall life satisfaction. They also found that SB during the week was positively associated with quality of life and posited that whether participants viewed PA positively or negatively might explain this association [60].

During the pandemic, studies in the population at-large indicate that decreased PA was a leading risk factor for depression [61]. Moreover, during the lockdown, increases in SB were associated with poorer mental health, and increases in PA were associated with better physical health [62]. Finally, self-isolation/quarantine was associated with higher depressive and anxiety symptoms, and individuals who changed their PA and SB during lockdown had greater physical and mental health [63]. With regard to ECE teachers specifically, a nationwide online survey on the COVID-19 impact, we reported that approximately 20% of ECE teachers experienced negative changes in their physical well-being during the pandemic [9]. Their greatest concerns related to health during the pandemic were weight gain, decreased PA, and increased SB.

PA is one of the most important and modifiable behaviors that can improve an ECE teacher’s physical health (e.g., BMI), along with mental health. Lessard and colleagues [34] reported that prior to the pandemic, ECE teachers had a higher prevalence of being overweight and obese. Whitaker et al. [38] indicated that several health risks are more prevalent in ECE teachers (i.e., Head Start staff) than in the national sample (e.g., more frequent unhealthy days, three or more physical health conditions). In addition, Linnan [36] reported that they have high levels of SB and can be inactive for nearly 9 h in the day. Decreased PA and increased SB during the pandemic may exacerbate existing health issues and social inequities for ECE teachers. However, the information for ECE teachers is scant.

Given the reciprocal and correlational nature of the interrelationships among PA, SB, physical and psychological well-being, life satisfaction, and the influence of changes brought on by the pandemic, a greater understanding of its impact on ECE teachers and their overall life satisfaction is imperative. Thus, this study aims at investigating associations among PA, SB, well-being (i.e., physical, psychological, and professional well-being), and life satisfaction among ECE teachers during the pandemic. Based on the interrelationships identified in previous studies, we developed and tested two competing mediational models. In the first model, we hypothesized that teachers’ well-being mediates the association of PA and SB with life satisfaction (see Figure 1). In our prior research [9], physical and psychological well-being played an important role in mediating the association of workplace resources and demands on professional well-being. However, based upon evidence in the literature on the role of PA and SB in one’s well-being and life satisfaction [64,65,66,67], we also tested the alternative hypothesis that PA and SB mediate the association between teachers’ well-being (i.e., physical, psychological, and professional well-being) and life satisfaction (see Figure 2). Our goal was to determine which of these hypothesized models best fit the data on ECE teachers during the early months of the pandemic (i.e., May to July in 2020).

## 2. Materials and Methods

### 2.1. Participants and Setting

A total of 1434 ECE teachers serving children aged between 0 and 5 (including Kindergarten) in 46 states in the United States completed the online survey from May to July in 2020, which is an early phase of the COVID-19 pandemic. The average age of the participants was 42 (Range *=* 17 to 80, *SD =* 12.03). The majority of teachers in the sample were women (98.3%). The participating teachers had diverse racial/ethnic and educational backgrounds. The overall racial/ethnic composition of the sample is similar to the population of ECE teachers nationally [68], with a somewhat higher percentage of Hispanic ECE teachers. The sample includes 60% White, 20% Hispanic, 14% Black, 4% American Indian or Alaska Native, 3% biracial, and 1% Asian. Of the participating teachers, 53% held a bachelor’s degree or higher, followed by some college but no degree (20%), associate degree (19%), and high school diploma or graduate equivalency (5%). The majority of teachers in the sample were fully paid (83%), but 12% were only partially paid, and 5% were not paid at all. Among those who were paid, the annual salary for more than half of the participating teachers was below USD 30,000. Furthermore, 15% of teachers were on welfare and received some form of public support such as Medicaid, food stamps, or childcare subsidies.

Participating teachers worked in Head Start programs (43%), childcare centers (34%), public schools (14%), family childcare homes (6%), and private schools (3%). They served infants and toddlers (24%), preschool or pre-K (38%), Kindergarten (6%), and children in multiage groups (31%). Of the 1234 ECE teachers in the sample, approximately 27% reported they were teaching in person, 37% were teaching online, and the remaining 36% were not teaching due to their centers being closed as a result of the pandemic.

### 2.2. Research Procedure and Analysis

Our interdisciplinary research team developed a 73-item online survey. It began with a description of the study and confidentiality statement and informed consent was acquired by the participants deciding to complete the survey. The first part of the survey collected information related to demographics, geographic location, socioeconomics, education, and work-related information. The survey collected information on physical well-being including PA, SB, psychological well-being, professional well-being, job demands, and life satisfaction during the pandemic. Portions of the survey consisted of previously validated scales and many items included text boxes for open-ended responses. More detailed descriptions and the psychometric properties of each measure are organized in Table 1.

#### 2.2.1. Measures of Physical Well-Being

Elements of the survey that collected information on teacher’s physical well-being included measures of general health condition, BMI, PA, SB, sleep, food security, and ergonomic pain. We measured PA and SB using elements of the International Physical Activity Questionnaire (IPAQ) Short Last 7 Days Self-administered format [72] and the SF-12 Health Survey Standard, Version 1 [69]. Hours of sleep were collected using a scale from 4 or less, 5, 6, 7, 8, 9, or 10 or more hours. Food security (including nutrition and eating habits) was measured using a modified version of The United States Department of Agriculture Short Form of the Food Security Survey Module [83]. Ergonomic pain was assessed using the modified version of the Work-Related Musculoskeletal Disorders Scale (WMDS) [74].

To measure PA and SB, direct survey questions asked about how many days and hours teachers spent in moderate to vigorous physical activities, and how much time they spent sitting on a weekday. The IPAQ is a frequently used measure of PA, with extensive reliability and validity studies, that spans 12 countries [72,73]. The short format captures PA in the last seven days with questions specific to vigorous PA and questions specific to moderate PA, along with how many days, as well as how many hours, in a week ECE teachers spent in each. The Federal Physical Activity Guidelines (PAG) and the ACSM diverge on their recommendations of vigorous PA each week. Federal PAG guidelines recommend 75 min/week and ACSM suggests 60 min/week, however, they both agree on 150 min/week of moderate PA. As such, we modified the survey to ask about moderate to vigorous PA, using 150 min/week as our threshold. “Insufficiently active” is defined by the U.S. Department of Health and Human Services Physical Activity Guidelines for Americans [84] as “doing some moderate or vigorous-intensity physical activity but less than 150 min a week.” In this study, we equate insufficiently active with being sedentary or engaging in prolonged bouts of sitting and we collected information on hours of sitting during a weekday.

#### 2.2.2. Measures of Physical Activity and Sedentary Behavior

The research team developed evidence-informed guidelines for assessing the frequency of PA and SB. We used 150 min per week (or 30 min five days a week) of moderate to vigorous PA every week as the threshold for meeting PA guidelines [84,85,86]. The available evidence for thresholds of both PA and SB is sparse. Although studies encourage the population to “move more and sit less” [85] and to break up prolonged sitting times with PA [86], few provide specific amounts of time. In a nationally representative survey of US adults over nine years, Du and colleagues [85] determined that adults sit an average of six hours per day. Only one study by Tremblay and colleagues [87] offers a specific metric, recommending “for adults aged 18 to 64 years and adults aged 65 years and older we recommend limiting sedentary time to 8 h or less,” yet they acknowledge that these are “strong recommendations based on low-to-very-low-quality evidence.” For survey data related to sitting times, if respondents provided exact estimates, we used that information in our data set. If they used a descriptive in the provided text boxes, such as “half the workday”, “some”, or “a fair amount”, we used six hours as a median [85]. We used eight hours or more if respondents used phrases such as “too much” or “almost all day,” or the equivalent per Tremblay et al. [87]. If responses were for a total week instead of a day (since the IPAQ asks for both), we used multiples of the preceding numbers (e.g., 30 h per week or six hours × 5 weekdays would be our average, and 40 h or above would be “too much”).

#### 2.2.3. Measures of Psychological Well-Being

Teachers’ psychological well-being was operationalized via ECE teachers’ perceptions of life satisfaction, depressive symptoms, stress, resiliency, and secondary trauma. Life satisfaction was gauged using the Satisfaction with Life Scale [75]. We assessed teacher depressive symptoms via the 10-item Center for Epidemiologic Studies of Depression Short Form (CES-D-10) [88]. Stress was measured using The Perceived Stress Scale (PSS) [79] and resiliency was measured using the Brief Resilience Scale [89]. Teachers’ secondary trauma was assessed using one of the subscales of the Professional Quality of Life Scale [90]. Finally, the SF-12 Health Survey Standard, Version 1 [69], mentioned previously as a measure of physical health also produces a mental health sub-scale. We calculated a simple, raw score for each of the SF-12 physical and mental health sub-scales. To preserve intact cases, sporadic missing data on the various items of the SF-12 were imputed via item correlation substitution (ICS). In ICS, a missing value for one item is replaced by the value of the measured item with which is it most highly correlated. Such a method of imputation is effective in scales with few response options and a low percentage of missing values [91]. In the case of the SF-12, this approach was robust from 2 but less than 3 missing values per case, which constituted 99% of all cases. Cases that had more than 3 missing values were ignored. More discussion of the handling of missing data for other study variables is described in more detail below in the data analysis section.

#### 2.2.4. Measures of Professional Well-Being and Job Demands

Professional well-being was assessed via two constructs: work commitment and intent to leave. Work commitment was measured using the Early Childhood Job Satisfaction Survey (ECJSS) [80]. Intent to leave the field/profession was measured via three survey items. Measures of job demands were assessed using The Job Content Questionnaire (JCQ) [92], which consists of three subscales: physical job demands, skill discretion, and decision authority.

After receiving Institutional Research Board (IRB) approval, we recruited ECE teachers to participate in the survey via various social media platforms (e.g., Facebook, Twitter) as well as e-mails to early childhood organizations, agencies, programs, and schools. As an incentive, we offered survey respondents the chance to win one of 120 USD 50 gift cards in a random drawing.

### 2.3. Data Analysis

To test our hypothesized models, we employed a structural equation modeling (SEM) approach via AMOS 25.0 using maximum likelihood estimation. Additionally, to adjust for any non-normality in our key variables, we generated bootstrapped estimates, standard errors, and bias-corrected confidence intervals using the AMOS program with a resampling size of 1000. In testing our model, we considered the following covariates as controls: (a) whether the teacher earned a bachelor’s degree or not, (b) teacher income, and (c) center status/teaching modality (open center, online/virtual, or closed center) at the time of the study. The age and ethnicity of teachers were not included as controls for reasons of parsimony; they exhibited no relationship to the endogenous variables in the models and had negligible impact on model fit. The effective sample size for our analysis was 1234 teachers.

Teachers in the sample who were missing scores on any one of the variables in the analysis (with exception of the SF-12) were removed prior to analysis (i.e., the values were not imputed). Variables in the dataset, including the SF-12, exhibited a general item non-response pattern [93]. This missing data pattern manifests as gaps in item response that appear to be randomly dispersed throughout the dataset (i.e., MCAR). In this case, we endeavored to determine whether or not there were significant differences between teachers who were missing scores for our measured variables and those who were not—in other words, could the data be assumed to be missing completely at random (MCAR)—which is needed to justify listwise deletion. To test this assumption, we conducted a series of Bonferroni corrected t-tests and chi-squared tests of independence between teachers who had values and those for whom they were missing for each variable. We found no significant differences between the groups with respect to our measured variables, and thus, listwise deletion was a justifiable missing data-handling approach (Enders, 2010). The reason for singling out the SF-12 specifically for imputation was due to random missingness patterns across all of the cases, which were highly unique across its 12 items. Of the teachers, 86% responded to all items on the SF-12, but the other cases were missing in many differing patterns, which resulted in a substantial number of lost cases when the missingness of the other study variables was taken into account. Once ICS was conducted on the SF-12, the number of lost cases dropped from over 500 down to just 200.

SEM construction began by first conducting confirmatory factor analyses (CFAs) of all candidate-latent measures of well-being. Our criteria for including latent measures in our final SEM was based upon factor loadings given that three-item (or less) CFAs are just- or under-identified and, therefore, provide no fit statistics. Factor loadings for all variables included in the measurement model were well above 0.5. This initial analysis yielded latent measures for use in the overall structural model of psychological well-being and physical well-being. Once CFAs were conducted and our latent measures were established, our overall modeling approach was primarily driven by our conceptual framework and prior research, but also included an empirical component for the purposes of testing direct paths as well as refining and settling on a final model.

We first tested our hypothesized model with all possible paths entered between these constructs. From here, we employed a model trimming approach [94] based upon the results of the saturated model where we eliminated non-significant paths step-by-step and examined our path estimates, as well as absolute and incremental fit statistics (see the Results for a more detailed discussion) retaining those paths that were theoretically justified and supported by empirical data. Though our model was hypothesized as fully mediational, at this stage, we tested direct paths from our exogenous variables to the endogenous variables in each mediational model. We retained those direct paths that were significant and improved model fit. Finally, we entered the aforementioned control variables (earned bachelor’s degree, teacher income, teaching modality) and tested their direct relationships to each of the outcome variables in our model. We retained those that improved model fit or were otherwise theoretically important, whether or not they were significant. Finally, as a final test of our framework, we tested the fit of our main alternative model against our hypothesized model specification. We were looking for a good-fitting model that also represented the more likely mediational reality of the relationship between physical activity, sedentary behavior, our three dimensions of well-being, and life satisfaction.

In assessing fit for the SEM analysis, we chose both measures of absolute (e.g., chi-square, RMSEA, pclose) and incremental fit (e.g., CFI, TLI). Established guidelines suggest that TLI and CFI statistics of 0.95 or higher indicate a good model fit; similarly, an RMSEA of less than 0.05 [95]. Furthermore, retaining the null of the pclose test, a test of whether or not RMSEA is “close-fitting” (i.e., RMSEA equal to 0.05), also indicates a good model fit. A non-significant chi-square value indicates a good overall fit; however, because it is well known to be sensitive to sample size (typically leading to a rejected null), we relied more heavily on our other chosen indices in assessing model fit. All variables in our SEM models have been standardized for ease of interpretation as effect sizes.

## 3. Results

Table 2 presents some basic descriptive statistics on our study sample and variables. Survey respondents produced a range of information with regard to income, educational level, and school status, along with metrics of overall life satisfaction, which included physical, psychological, and professional well-being. Nearly half of the ECE teachers reported an annual income between USD 20,000 and 30,000 (27.4%) and USD 30,000 and 40,000 (22.4%), and nearly equal percentages occupied the low and high ends of the range, with 5.8% making below USD 10,000 and 5.5% making above USD 70,000. Respondents had a mean life satisfaction score of 24.7, reflecting overall “slight satisfaction.”

The aggregate score of the full sample on the SF-12 physical health subscale was 16.47 (lowest possible score of 6 and highest of 20) [96] (p. 383), and their overall intensity of reported ergonomic pain (with a maximum of 20) was 3.50. They spent on average 6.5 h a day being sedentary, with just over a third (39%) meeting the recommended 150 min of moderate-intensity PA per week. Mean BMI was 31.5, which classifies as obese, with 0.8% of participants classifying as underweight, 22.0% were normal BMI, 27.0% were in the overweight category, and 50.2% in the obese category [97]. The range of BMIs was from 12.73 to 80.69, indicating that, if height and weight were accurately self-reported, some respondents weighed over 500 pounds. From a psychological well-being standpoint, survey respondents had a mean score of 16.6 on the SF-12 psychological health subscale (MCS) (lowest possible score of 6 and highest of 27), their mean perceived stress score was 25.77, indicating the upper end of “moderate stress” (Cohen et al., 1983) and their mean depressive symptoms were 8.47 (a score of 10 or higher indicates depression) (Eaton et al., 2004). The teachers’ mean work commitment score was 8.52 (0 = low work commitment, 10 = high work commitment) [80].

Finally, to provide a sense of the underlying relationships between our key variables prior to the SEM analysis, zero-order correlations among key variables are presented in Table 3. Of note are the moderate relationships between our well-being measures as well as these well-being measures and our outcome life-satisfaction. Additionally notable, is the lack of a significant relationship between PA (more than 150 h of moderate exercise per week) and SB (number of sedentary hours per day), *r* = 02, *p* = 0.92).

### Empirical Tests of Competing Mediation Models

The overarching inquiry of this study was whether PA or aspects of well-being serve as the better mediator of the associations between PA and ECE teachers’ life satisfaction. Our final well-being mediation model is displayed in Figure 3. The empirical test of this mediation model revealed very good, close model fit, χ^2^ (54) = 188.53, *p* < 0.001, CFI = 0.963, TLI = 0.946, RMSEA = 0.045, 90% CI (0.038, 0.052), pclose = 0.877 (ns), according to established SEM guidelines [98]. Overall, these findings suggest that the latent and observed measures and the relationships between them in our recursive, fitted model matched the covariance structure of the data well. Our final model explained approximately 40% of the variance in life satisfaction, 26% of the variance in psychological well-being, and approximately 10% of the variance in professional well-being.

With respect to the well-being mediation model, there were several key findings. First is that the three aspects of teacher well-being (physical, psychological, and professional) mediated the relationship between PA, SB, and life satisfaction. There were two main indirect paths from PA (i.e., more than 150 min of moderate PA per week) to life satisfaction. The weaker of the two was through all three well-being measures (physical, psychological, and professional well-being; indirect β = 0.002). The stronger of the two flowed through only physical and psychological well-being; indirect β = 0.053. The total indirect effects of PA on life satisfaction were significant, β = 0.055, *SE* = 0.011, *p* < 0.001, 95% CI (0.038, 0.079). The total indirect effects of SB on life satisfaction through physical, psychological, and professional well-being were also statistically significant; indirect β = −0.065, *SE* = 0.019, *p* < 0.01, 95% CI (−0.101, −0.029).

Of note was the indirect effect of physical well-being on life satisfaction through psychological well-being, which was over three-tenths of a standard deviation, β = 0.307, *SE* = 0.024, *p* < 0.001, 95% CI (0.262, 0.354), as well as the direct effect of psychological well-being on life satisfaction, which was well over a half a standard deviation, β = 0.589, *SE* = 0.025, *p* < 0.001, 95% CI (0.540, 0.637). Not surprisingly, more than 150 min a week of moderate PA was directly and positively associated with physical well-being, β = 0.178, *SE* = 0.038, *p* < 0.001, 95% CI (0.126, 0.243), and sedentary behavior was negatively associated with physical, β = −0.068, *SE* = 0.032, *p* < 0.05, 95% CI (−0.134, −0.009), and psychological well-being, β = −0.080, *SE* = 0.029, *p* < 0.01, 95% CI (−0.136, −0.022), respectively.

Finally, our empirical test of the PA mediation model is displayed in Figure 4. This model had good, close fit, but was slightly inferior in fit to the well-being mediation model, χ^2^ = 195.47, df = 55, *p* < 0.001, TLI = 0.945, CFI = 0.961, and RMSEA = 0.046, 90% CI [0.039–0.053], pclose = 0.851. Furthermore, an empirical test of a potential PA mediation model revealed, in fact, no mediation, due to non-statistically significant paths from SB to life satisfaction and from PA to life satisfaction, direct β = −0.018, *SE* = 0.025, *p =* 0.481, 95% CI (−0.071, 0.030) for SB; β = 0.037, *SE* = 0.022, *p =* 0.116, 95% CI (−0.009, 0.075) for PA respectively.

## 4. Discussion

The present study investigated the associations of PA and SB with ECE teachers’ well-being and life satisfaction during the early phase of the COVID-19 pandemic in 2020. We explored two mechanisms through which PA, SB, and well-being are associated with ECE teachers’ life satisfaction: one with PA and SB as mediators and the other with teachers’ well-being as a mediator. The overall conclusion of this analysis is that, while both models fit well, the model where teacher well-being was the mediator was superior. This is not only because it fit slightly better, but because it represented a mediation, whereas the model of PA was not technically a mediation, having non-significant paths from well-being to life satisfaction. We unpack these results more below, beginning with the well-being mediation model.

In addition to informing our model development, data from the survey produced a wealth of information about various aspects of ECE teachers’ well-being, their levels of PA, SB, and overall life satisfaction during the COVID-19 pandemic. Respondents had a mean life satisfaction score that qualified as slight satisfaction, they experienced moderate stress, and, collectively, were approaching the threshold for depression yet still reflected moderate-to-high work commitment. Over half of the ECE teachers in our sample were classified as obese, which is similar to a finding by the authors [8] where we found a 55% prevalence of obesity in a pre-pandemic study of ECE teachers. Further, only 39% of teachers met the recommended 150 min of moderate PA per week, which is higher than the 20% reported for American adults aged between 18 and 65 years old by the Office of Disease Prevention and Health Promotion [84]. They spent, on average, 6.5 h a day being sedentary, which, when factoring in approximately 8 h of sleep, falls within the range from 50–60% of SB seen in the US adult population reported by Healy and colleagues [99]. Patterson et al. [18] suggest that from 6–8 h of sitting is a clear threshold of disease risk, and the US Department of Health and Human Services (HHS) [84] also reports a strong relationship between time in SB and the risk of all-cause mortality and cardiovascular disease mortality in adults. This said, the HHS also states that the degree of risk related to SB is dependent upon the amount of moderate-to-vigorous PA performed, which hints at the complex relationship between PA, SB, overall well-being, and life satisfaction revealed in our modeling.

The empirical test of our competing mediation models found the superior model was the model where teacher well-being mediated the association between PA and SB and life satisfaction compared to the model where PA and SB mediated the association between teacher well-being and life satisfaction. These findings add to the existing literature that high PA and low SB are associated with life satisfaction [52,58]. In our final model, all three aspects of teacher well-being (physical, psychological, and professional) mediated the relationship between PA and SB to life satisfaction. Previous research utilizing a multidimensional approach to understanding well-being found well-being explained substantial variation in key outcomes such as job satisfaction (58.6%), organizational commitment (40.6%), and life satisfaction (42.3%) [47]. Although two of these outcomes were not measured in our study, we found our final model explained a similar amount of variation in life satisfaction for our sample of ECE teachers. Life satisfaction is impacted by a complex set of environmental, individual, and social factors. Identifying implementation strategies that focus on improving PA and SB as well as well-being (physical, psychological, and professional) will be important in improving this outcome. As previous research has found the type of PA (e.g., household, leisure, transportation) to be related to quality of life but not overall life satisfaction, and because minimal research has explored factors of well-being in ECE teachers, additional research examining more specific aspects of PA and its relationship to well-being and life satisfaction is warranted [29].

We observed two main indirect paths from PA (i.e., more than 150 min of moderate PA per week) to life satisfaction. The weaker of the two was through all three aspects of well-being measures (physical, psychological, and professional) and the stronger flowed through only physical and psychological well-being. The total indirect effects of PA on life satisfaction were significant as were the total indirect effects of SB on life satisfaction through physical, psychological, and professional well-being. These findings support research from others [52,58] in that PA and SB are associated with higher life satisfaction, although effects measured in these studies were limited in younger adults [58]. These differences suggest there may be developmental differences throughout the life course or increase the importance of engaging in PA through middle and late adulthood, which has implications for the development of targeted interventions. Furthermore, the direct and indirect relationships found in this study between PA and SB and psychological well-being emphasize the importance of the role of PA on mental health and depression [55,56,60].

Additionally observed, was a medium effect of the indirect path of physical well-being on life satisfaction through psychological well-being. An even larger effect was found in a direct path of psychological well-being on life satisfaction for this population, confirming the findings of Kardas and colleagues [49] who found that life satisfaction, along with gratitude, optimism, and hope, accounted for 51% of the variance in psychological well-being. The literature is unclear on the direction of these paths with regard to whether psychological well-being influences life satisfaction or whether life satisfaction influences psychological well-being. Our model confirmed that 150 min a week of moderate PA for ECE teachers was directly and positively associated with physical well-being, whereas SB was negatively associated with physical and psychological well-being. Thus, PA and SB remain modifiable supportive factors for ECE teachers and can improve overall physical well-being, decrease obesity, and hold lifestyle diseases at bay [23,53,54]. The increase in SB could also be partially explained within the context of the pandemic where almost half of the EC facilities closed, and equal percentages remained open or taught online leaving teachers prone to increased SB while teaching online or in their home environment and with limited to no access to fitness activities or facilities.

Our empirical test of the PA mediation model had a good, close fit, but was slightly inferior in fit to the well-being mediation model. Most importantly, however, the test of this model revealed no actual mediation. We surmise that this is due to the complex nature of PA on enhancing physical and psychological well-being and, ultimately, life satisfaction. Further, this suggests that PA, in the absence of enhanced psychological well-being, may have a limited impact on life satisfaction. These findings highlight the important role both PA and SB play in their well-being and life satisfaction. The findings are particularly important for ECE teachers, as previous research has found that these teachers have many health and well-being-related risks (e.g., obesity, depressive symptoms, perceived stress, [9]. Moreover, physically active ECE teachers would serve as role models and motivate children to be physically active [20,21,22].

### Limitations and Recommendations

Caution is warranted in the application or generalization of our findings, however. First, it is important to note that our data were collected during the early months of the COVID-19 pandemic when stress and life disruptions were quite high and disproportionately impacted vulnerable populations, including ECE teachers [100]. Thus, it is likely that the typical associations we might see between PA, SB, well-being, and life satisfaction were also disrupted. Second, PA and SB were assessed with singular, self-reported items using a gross assessment of time. It is plausible and likely that more sensitive measures of movement, behaviors, and context would strengthen the findings and understanding of the complex nature of well-being. Indeed, our reliance on the self-reporting of information using the survey format was a limitation. Although self-report estimates of height are usually within 0.1-inch accuracy, self-report of weight is typically underestimated by 4.6 pounds [101], and substantive variability exists in self-reporting of PA [102]. Moreover, given the variability in recommendations for PA and the dearth of suggestions for SB, the benchmarks we used to determine thresholds—although evidence-informed—are somewhat tenuous. Further, information on PA and SB in our survey was assessed using estimates of time.

## 5. Conclusions

The overall conclusion of this analysis is that, while both models fit well, the model where teacher well-being was the mediator flowing through only physical and psychological well-being was superior. Of the 1234 ECE teachers who completed the online survey, over three quarters were overweight or obese, and just above a third of teachers met the recommended 150 min of moderate PA per week, which we found was positively associated with physical well-being and sedentary behavior was negatively associated with physical and psychological well-being. They have slight life satisfaction, experience moderate stress, and collectively, are approaching the threshold for depression yet still reflect moderate-to-high work commitment. Future studies to understand this somewhat counterintuitive finding is warranted. In our final model, all three aspects of teacher well-being (physical, psychological, and professional) mediated the relationship between PA and SB to life satisfaction. Since life satisfaction is influenced by a complex set of factors beyond those that we examined in this study, further research is warranted to determine the role that other factors, including environmental, individual, and social factors, can have on it. Given our findings of the direct and indirect relationships between PA, SB, and psychological well-being, as the pandemic subsides, we remain optimistic that PA and SB will remain modifiable risk factors for ECE teachers to improve their overall physical and psychological well-being, along with their life satisfaction.

## Figures and Tables

**Figure 1 ijerph-18-09430-f001:**
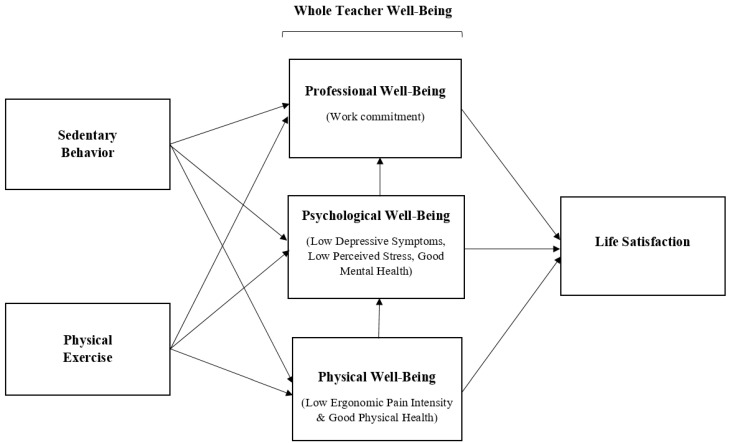
Well-being Mediation Model of Physical Activity and Life Satisfaction.

**Figure 2 ijerph-18-09430-f002:**
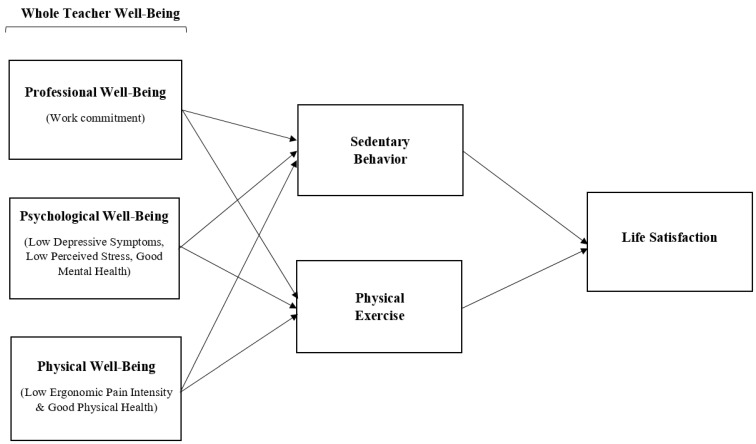
Physical Activity Mediation Model of Well-being and Life Satisfaction.

**Figure 3 ijerph-18-09430-f003:**
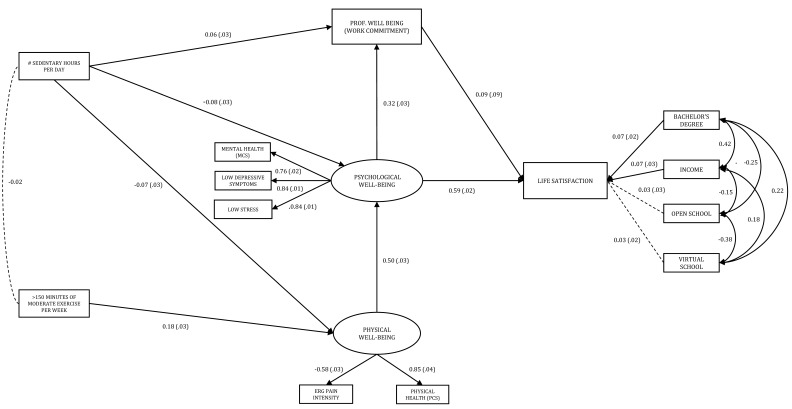
Empirical Test of Well-Being Mediation Model. Note. Bootstrapped standardized estimates and standard errors reported (resampling size = 1000), standard errors in parentheses. The regression weights for PCS and low stress were constrained to 1 for analysis. All paths significant *p* < 0.01 except sedentary hours per day to professional well-being, *p* < 0.05. Dotted paths/covariances non-significant. Fit statistics: χ^2^ = 188.53, df = 54, *p* < 0.001, TLI = 0.946, CFI = 0.963, and RMSEA = 0.045, 90% CI [0.038–0.052], pclose = 0.877.

**Figure 4 ijerph-18-09430-f004:**
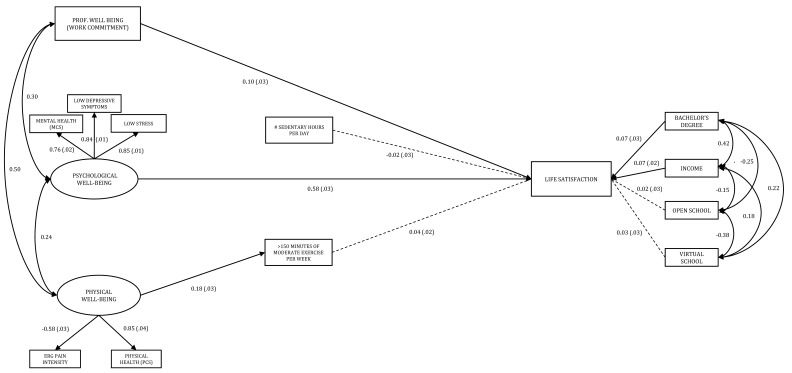
Empirical Test of Physical Activity Mediation Model. Note. Bootstrapped standardized estimates and standard errors reported (resampling size = 1000), standard errors in parentheses. The regression weights of PCS and low depressive symptoms were constrained to 1 for analysis. All paths significant *p* < 0.01. Dotted paths/covariances non-significant. Fit statistics: χ^2^ = 195.47, df = 55, *p* < 0.001, TLI = 0.945, CFI = 0.961, and RMSEA = 0.046, 90% CI [0.039–0.053], pclose = 0.851.

**Table 1 ijerph-18-09430-t001:** Measures Used in the Current Study.

	Study Variables	Instruments	Instrument Characteristics	Psychometric Properties
**Physical Well-being**	Overall Health Status	SF-12 Health Survey Standard, Version 1 [69]	The SF-12 is a shortened version of the SF-36 with questions in 8 domains. The survey includes a Physical Component Score (PCS) subscale.	Reliability α = 0.78–0.85 [70] Cronbach’s α = 0.83 [71]
	Physical Activity (PA)	International Physical Activity Questionnaire (IPAQ) Short Last 7 Days format [72]	7-item self-administered questionnaire providing an estimate of physical activity and sedentary behavior over the last seven days. Used with adults aged 15–69 years old.	Reliability α = 0.80 Criterion Validity Spearman’s ρ = 0.30 [73]
	Days/hours spent on PA	Questionnaire	Ascertains days and hours spent in moderate to vigorous physical activities (150 min/week used as threshold) on a weekly basis	
	SB/Hours of Sitting	Questionnaire	Time spent on sedentary activities in past four weeks (none of the time, a little of the time, some of the time, or a good bit of the time)	
	Ergonomic Pain	Modified Work-Related Musculoskeletal Disorders scale [74]	Five binary items asking about experienced pain in neck, back, shoulder, knee, and other. Total score combined all items.	Cronbach’s α = 0.90; Test-retest reliability Pearson r > 0.75 [74]
**Psychological Well-being**	Life Satisfaction	Satisfaction with Life Scale [75]	Scale consists of 5 items measuring global cognitive judgments of a person’s life satisfaction ranging from 1 (strongly disagree) to 7 (strongly disagree).	Test-retest correlation coefficient α = 0.82 coefficient α = 0.87 [75]
	Depressive Symptoms	Center for Epidemiologic Studies of Depression Short Form (CES-D-10) [76]	10-item screening test on frequency of symptoms in the past week on a scale of 0 (not at all or less than 1 day) to 3 (5–7 days). Scores equal to or above 10 are considered to indicate a screen of depression	CES-D-10; Cronbach α = 0.65–0.91 [77,78]
	Stress	The Perceived Stress Scale (PSS) [79]	Questions about current levels of experienced stress ranging from 1 (rarely/never) to 5 (very often). The PSS is a predictor of depressive and physical symptomatology (Cohen et al., 1983)	PSS; Cronbach α = 0.84; test-retest reliability Pearson *r* = 0.85 [79]
**Professional Well-being**	Work Commitment	Early Childhood Job Satisfaction Survey (ECJSS) [80]	Ten questions (true/false) exploring factors related to work satisfaction and commitment. Scores range from 0 (low) to 10 (high) levels of work commitment.	Overall consistency for ECJSS: α = 0.89 [81]; Internal consistency reliability α = 0.80 [82]

**Table 2 ijerph-18-09430-t002:** Descriptive Statistics for Study Variables (*n* = 1234).

Categories	Percentage OR Mean (Range OR SD)
Teacher Characteristics	
Held Bachelor’s Degree	0.55 (0.49)
Income	3.80 (1.70)
USD 10 k or less	5.8%
USD 10,001 to USD 20 k	15.9%
USD 20,001 to USD 30 k	27.4%
USD 30,001 to USD 40 k	22.4%
USD 40,001 to USD 50 k	14.4%
USD 50,001 to USD 60 k	5.7%
USD 60,001 to USD 70 k	3.0%
USD 70 k and up	5.5%
Teacher School Open During Pandemic	0.28 (0.45)
Teacher School Online Learning During Pandemic	0.27 (0.44)
Teacher School Closed During Pandemic	0.45 (0.49)
Life Satisfaction	24.74 (5–35)
Psychological Well-being	
Perceived Stress	25.77 (10–47)
Depressive Symptoms	8.47 (0–30)
Mental Health Scale (MCS, SF-12)	16.64 (2–23)
Professional Well-being	
Work Commitment	8.52 (1–10)
Physical Well-being	
Ergonomic Pain Intensity ª	3.50 (0–19)
Physical Health Scale (PCS, SF-12)	16.47 (4–20)
Physical Activity	
Sedentary hours per day	6.52 (3.41)
More than 150 min of moderate physical activity per week	0.39 (0.49)

*Note.* ª Is an indicator of the product of both number of affected areas (max of 5 areas) and severity of pain (from 0 = no pain to 4 = unbearable pain), for a max range of 20.

**Table 3 ijerph-18-09430-t003:** Zero-Order Correlations for Study Variables.

Measure	1	2	3	4	5	6	7	8	9	10	11	12
1. Life satisfaction	-----											
2. Mental health (MCS)	0.44 **	-----										
3. Low depressive symptoms	0.52 **	0.62 **	-----									
4. Low personal stress	0.51 **	0.63 **	0.71 **	-----								
5. Physical health (PCS)	0.29 **	0.38 **	0.33 **	0.30 **	-----							
6. Ergonomic pain intensity	−0.20 **	−0.29 **	−0.28 **	−0.23 **	−0.51 **	-----						
7. Work commitment	0.26 **	0.21 **	0.25 **	0.27 **	−0.20 **	−0.17 **	-----					
8. # Sedentary hours/day	−0.04 *	−0.12 **	−0.07 **	−0.05 *	−0.04	0.09 **	0.02	-----				
9. >150 min mod. Exercise/wk	0.10 **	0.13 **	0.08 **	0.07 **	0.15 *	−0.07 **	0.00	0.02	-----			
10. Income	0.09 **	0.02	0.03	0.03	0.14 **	0.01	0.03	−0.07 **	0.07 **	-----		
11. Bachelor’s degree	0.08 **	0.02	0.01	0.02	0.07 **	0.00	−0.05 *	0.11 **	0.06 *	0.42 **	-----	
12. Open school	−0.06 *	0.11 **	−0.03	−0.04	0.09 **	0.07 **	−0.06 **	−0.10 **	−0.02	−0.15 **	−0.25 **	-----
13. Virtual school	0.04	−0.01	−0.02	0.02	0.04 *	−0.02	0.00	0.08 **	−0.01	0.17 **	0.22 **	−0.40 **

** *p* < 0.01, * *p* < 0.05. *Note*. Non-parametric correlational analysis used for income, bachelor’s degree, 150 min exercise, online and virtual school. Ergonomic pain intensity is summed products of each point of pain and reported severity.

## Data Availability

Data supporting reported results can be accessed by communicating by email with the corresponding author.

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
