# Peer review of "Physical Activity, Physical Well-Being, and Psychological Well-Being: Associations with Life Satisfaction during the COVID-19 Pandemic among Early Childhood Educators"

_ijerph, 2021, doi:10.3390/ijerph18189430_

Round 1
Reviewer 1 Report
Dear Authors, thank you for a quite interesting study regarding a current topic: the predictors of life satisfaction during the COVID-19 pandemic.
You examined the association of physical activity and sedentary behaviours with the well-being of early childhood educators by exploring and comparing two different models with different mediators in a quite large cohort.
I have some comments to add to the different sections:
Abstract: Appropriate.
Introduction: Appropriate.
Methods: This section is appropriate. Just some brackets are missing (e.g., ... Center for Epidemiological Studies of Depression Short Form (CED-D-10; or The Perceived Stress Scale (PSS; and... .
Results: The results section contains parts of the discussion/conclusion (The overall conclusion of this analysis is that, while both models fit well, the model where teachers ... just beneath table 3). This should definitely not be part of the results section.
Discussion: This section is appropriate and the results are embedded in current evidence.
Conclusion: The conclusion shows redundances of the results section (e.g. first two sentences, last sentence of the first paragraph, first very long sentence of the second paragraph). Please specifiy the conclusion. Furthermore, there are no ideas and approaches regarding future studies named. Could you add any ideas for possible interventions or further research needed to construct suitable interventions?
Referencens: Appropriate.
Tables: All tables are understandable, but not consistent in style. In all tables, the abbreviations should be explained (MCS, SF-12), they should be understandable without reading the text.
Figures: Fig. 1 is really nice. Fig. 2 and 3 are okay,Fig. 4 and 5 are really small and barely readable. Is there any possibility to improve this?
All the best for your future work.
Author Response
Thank you so much for your keen eye and helpful suggestions.

Reviewer 2 Report
Thank you for this interesting topic and a huge study with many parameters. I find it is very interesting and you did a good job!
I just found several spaces after end of sentence in text during the document- maybe this should be looked up. On p.2 a iterature is in parentheses otherweise always inidivates with superscript (Ren, 2020).
Fig.2 and 3 the arrows are not well visible when many arrows are coming together.
Page 6: "The average age of the participatns was 42 - I would put in standard deviation afterwards.
Page 6 last sentence: I am missing a view percentages ;-) (27, 37 and 36 % is not 100%).
Page 7: Days/ hours spent on PA. This is an own questionnaire. How did you describe what is moderate to vigorous PA? (Also not comprehensible for me in the last sentence of p.9)
Page 9: Bracket missing after (BMI...)
Page 10:..."half the work day",...etc. did you distinguish between half- time and full - time employees when calculating the hours?
There is an underlindes line on p 10- I guess this is not really wanted ;-)
Author Response
Thank you for your keen eye and helpful suggestions.

Reviewer 3 Report
General comments:
Thank you for the opportunity for reviewing this manuscript. This study investigated the association between physical activity, sedentary behaviour and well-being with life satisfaction among early care and education teachers during the pandemic. In general, the manuscript is well written, but there are some concerns about the methods used in this study, particularly statistical analyses.
[Proposed models (Figures 2 and 3) vs. analytical models (Figures 4 and 5)]
First of all, Figures 4 and 5 are very fuzzy. It would be great if the authors provide figures with better resolution. Without clear figures, it is impossible to assess these figures and associated values in the final models. Please also include what solid and dotted lines mean, and any abbreviations (if any) used in the figures.
When comparing the prosed models and the analytical models, there are slightly different each other (e.g., no direct path from physical well-being to life satisfaction [Figure 4]; and direct paths were created between 3 well-being indicators and life satisfaction [Figure 5]). Also, it seems that not all of the measures described in Table 1 were included in the final models. In the results, the modification process (or ‘a model trimming approach’ according to the authors) was not clearly reported and explained (which paths were non-significant, and which paths were eliminated from each model). Without the information on model modification, it is hard to assess and/or compare two models as their direct/indirect paths might be different each other. Also more analyses were performed, more chances to get significant results. It may be also ideal to use different subsets of the sample (e.g., randomly dividing the sample into two [n = 617]) to test each model (rather than using the same sample [n=1234] across two models).
Regarding the control variables, why age and ethnicity were not included in the final models? Standardisation of the results were not mentioned in the data analysis – please explain if effect sizes were standardised in the model.
[Materials and Methods]
This section includes extra information which should be reported in the results section or which repeated the same information shown in Table 1. For example, most information in the participants and setting on page 6 can move to the results section. Is the information included in this section related to the sample of 1434 ECE teachers? If so, was the data changed when only included 1234 ECE teachers who were included in the final models? Age, sex and ethnicity should be reported in Table 2 too.
Physical activity measures included in the physical well-being in Table 1 is rather confusing – Based on the proposed models, physical activity and sedentary behaviour are separated from the physical well-being. However, in Table 1, these two measures were included as part of the physical well-being. Please present physical activity and sedentary behaviour measures separately from the physical well-being measures in this table. If physical activity and sedentary behaviour measures were included in physical well-being as well as exposure or mediating variables, there is an issue of multicollinearity in the models.
In terms of physical activity and sedentary behaviour measures, it says “For survey data related to sitting times, if respondents provided exact estimates, we used that information in our data set. If they used a descriptive in the provided text boxes, such as ‘half the work day’ or ‘some’ or ‘a fair amount’, we used six hours as a median. …” – how many responses (or percentages) were based on descriptive answers? This may impact on the results (accuracy of reporting in sedentary behaviour).
Imputation for missing data only for the SF-12 was reported – what about the other survey data? Also how item correlation substitution (ICS) works if there are more than one missing values in the SF-12?
Minor comments:
[Title]
Please use ‘associations’ rather than ‘predictors’ as this study used cross-sectional data only.
[Abstract]
A sentence describing the study aim is lengthy – please break it into two. Also please capture the overall study aim – for example, it says “the associations of physical activity and sedentary behaviours with teacher wellbeing and life satisfaction”. In the main manuscript, however it says “associations among PA, SB, wellbeing (i.e., physical, psychological and professional well-being), and life satisfaction among ECE teachers during the pandemic (page 5)”.
‘the superior model’ – comparison is not the purpose of this study. In addition, these two models should not be compared without clearly describing what modification was made between the two models and if they are comparable or not.
[Introduction]
‘early in the pandemic’ – if possible, please include month(s) as each country had slightly different lockdown restrictions in place, and so as the timing of lockdown. Similarly, please specify the month(s) in the methods (page 6), rather than ‘late Spring to mid-summer’ for the international audience.
There are a couple of places (e.g., page 2, the last sentence before Benefit of PA for ECE Teachers) where SB was excluded as a mediator or the associations (only PA was highlighted).
Page 3: Professional well-being was not explained in relation to Figure 1. This paragraph mainly focuses on explaining Figure 1, but life satisfaction is not included in Figure 1.
Pages 4-5: A paragraph related to the pandemic was briefly mentioned in the previous paragraph. Please avoid repetition, and summarise the section where possible.
[Materials and Methods]
Page 7: “It then gathered data specific to COVID-19 that included personal experiences with infection and how the pandemic influenced the teachers’ work and home life” – was this question item(s) used in this study?
Page 9 (Measures of Physical Well-being): Please avoid repeating the same information shown in Table 1. Table 1 nicely summarised each measure. Instead, the information which do not included in the Table 1 should be described in text (e.g., number of items, how score was summarised for composite items like SF12, IPAQ, Food Security etc.). This information will be helpful when interpreting results from Table 2 (when presenting %s or means). In relation to Table 2, measures for ‘teacher school open during pandemic’, ‘teacher school online learning during pandemic’, and ‘teacher school closed during pandemic’ was not clearly described in the methods. Also why ‘held bachelor’s degree’ and ‘more than 150 min of moderate physical activity per week’ were shown as mean rather than %? (Please use ‘physical activity’ rather than ‘exercise’ throughout the manuscript as this study examined PA rather than exercise only. Please describe all abbreviations used in this table in the footnote.)
[Results]
Page 13: Interpretation of model fit indices should be described in the methods (data analysis) rather than the results section.
Table 3: It is not clear how results from Table 3 were used in the current analyses. Based on this table, life satisfaction and wellbeing measures were more strongly correlated than sedentary behaviour and physical activity. This correlation table may explain the reason why the physical activity and sedentary behaviour mediation model did not perform well compared to well-being mediation model.
Page 13 (paragraph after Table 3): Based on the study aim stated in the introduction, the comparison between the two models were not the main purpose. It is also difficult to compare them if any modifications were made. As mentioned above, please fully describe what kind of modification was made in each model and how these modifications improved each model fit accordingly (to reach the final models).
[Discussion]
Page 15: The study aim included here slightly different from what it was stated in the introduction (but the same as the abstract). SB was excluded as a mediator.
Page 16: “the weaker of the two was through all three aspects of well-being measures and the stronger flowed through only physical and psychological well-being” – to compare these values directly, standardisation is required. However, standardisation was not mentioned in the methods.
Page 17: “Most importantly, however, the test of this model revealed no actual mediation” – there is a mediating effect, but not statistically significant?
[Conclusions]
Please shorten the conclusions by removing the first paragraph (except the last sentence). The conclusion is not abstract, so no need to state the study aim and methods, but key findings and their applications/future directions.
Author Response
Thank you for your feedback and suggestions.

Round 2
Reviewer 3 Report
The authors responded to the reviewer's comment sufficiently, and improved the manuscript. No further comments are provided.